# Functionalized Halloysite Nanotubes as Potential Drug Carriers

**DOI:** 10.3390/jfb14030167

**Published:** 2023-03-21

**Authors:** Ewa Stodolak-Zych, Alicja Rapacz-Kmita, Marcin Gajek, Agnieszka Różycka, Magdalena Dudek, Stanisława Kluska

**Affiliations:** 1Faculty of Materials Science and Ceramics, AGH University of Science and Technology, al. A. Mickiewicza 30, 30-059 Krakow, Poland; 2Faculty of Energy and Fuels, AGH University of Science and Technology, al. A. Mickiewicza 30, 30-059 Krakow, Poland

**Keywords:** halloysite nanotubes, gentamicin, functionalization, drug carrier, antibacterial activity

## Abstract

The aim of the work was to examine the possibility of using modified halloysite nanotubes as a gentamicin carrier and to determine the usefulness of the modification in terms of the effect on the amount of the drug attached, its release time, but also on the biocidal properties of the carriers. In order to fully examine the halloysite in terms of the possibility of gentamicin incorporating, a number of modifications of the native halloysite were carried out prior to gentamicin intercalation with the use of sodium alkali, sulfuric and phosphoric acids, curcumin and the process of delamination of nanotubes (expanded halloysite) with ammonium persulfate in sulfuric acid. Gentamicin was added to unmodified and modified halloysite in an amount corresponding to the cation exchange capacity of pure halloysite from the Polish Dunino deposit, which was the reference sample for all modified carriers. The obtained materials were tested to determine the effect of surface modification and their interaction with the introduced antibiotic on the biological activity of the carrier, kinetics of drug release, as well as on the antibacterial activity against *Escherichia coli* Gram-negative bacteria (reference strain). For all materials, structural changes were examined using infrared spectroscopy (FTIR) and X-ray diffraction (XRD); thermal differential scanning calorimetry with thermogravimetric analysis (DSC/TG) was performed as well. The samples were also observed for morphological changes after modification and drug activation by transmission electron microscopy (TEM). The conducted tests clearly show that all samples of halloysite intercalated with gentamicin showed high antibacterial activity, with the highest antibacterial activity for the sample modified with sodium hydroxide and intercalated with the drug. It was found that the type of halloysite surface modification has a significant effect on the amount of gentamicin intercalated and then released into the surrounding environment but does not significantly affect its ability to further influence drug release over time. The highest amount of drug released among all intercalated samples was recorded for halloysite modified with ammonium persulfate (real loading efficiency above 11%), for which high antibacterial activity was found after surface modification, before drug intercalation. It is also worth noting that intrinsic antibacterial activity was found for non-drug-intercalated materials after surface functionalization with phosphoric acid (V) and ammonium persulfate in the presence of sulfuric acid (V).

## 1. Introduction

Clay minerals have the ability to incorporate large amounts of various substances, such as macromolecules, drugs, DNA, or proteins [1,2,3,4,5,6], due to their excellent sorption properties and are widely used in the pharmaceutical industry. It has been documented that they can gradually release the incorporated drug in the environment of a living organism, which is particularly useful in the design of drug delivery systems with its controlled release [2,4]. The possibility of designing such systems combined with the carrier’s inertness to the human body creates a huge field for searching for the best solutions, and one such solution may be the not fully explored possibility of using halloysite. Halloysite is one of the safest clays for biological applications [7,8], which, in addition to biocompatibility and low toxicity [9], has a unique tubular structure, different from clay minerals, which creates an additional possibility of introducing the drug into the lumen of the nanotube, and not just adsorbing it on its surface. The main purpose of incorporating drug particles into halloysite is to extend the time of their release in the living organism possibly even up to several hours, while maintaining the minimum dose of the drug [10]. This additionally slows down the release of the active substance and creates a special potential for using this clay as a drug carrier [7,11]. There are numerous ways to introduce the drug into the halloysite, and one of the simplest methods is to mix the drug with the nanotubes in an aqueous environment, ensuring that the amount of drug added is consistent with the calculated ion exchange capacity (CEC) of the halloysite. Nevertheless, there is an increasing need to search for new, more effective methods of introducing the drug into an inexpensive, inert, and widely available mineral, and the literature on the subject contains the results of research on several different antibiotics in terms of their introduction into halloysite nanotubes, namely: tetracycline, ciprofloxacin and ofloxacin [12], which belong to the group of cationic drugs [2,4,5,6]. However, without chemical modifications of the carrier before the drug incorporation, the substance entrapped inside the nanotubes and adsorbed on the surface is desorbed and released usually within 10 to 20 h in a liquid environment [6]. Therefore, additional modification of halloysite before drug introduction seems to be a promising way to extend the time of sustained drug release. This is self-evident, as the specific structure of halloysite nanotubes (inner layer of aluminum hydroxide and outer layer of silica) allows for a selective surface interaction that can be modified by expanding the interior of the nanotube or by increasing its porosity. For this purpose, clay minerals are subjected to various modifications to increase their “loading capacity”, and because the possibilities of surface modification are almost countless, it has resulted in numerous research studies in this field [13,14,15]. Bonding of the active substance to the halloysite structure can take place through electrostatic interactions, as the drug molecules usually have a positive surface charge, while the surface of the halloysite is negatively charged [16,17]. The kinetics of drug release and the ability to bond to it are additionally affected by surface functionalization, especially alkaline, where OH^−^ groups are bound on the surface of the nanotube, increasing the value of the surface charge. Acid modification of the carrier, modification with organic substances or thermal treatment are also possible; however, the results of some studies show that the amount of bound active substance is lower than theoretical, resulting from the CEC value [4]. Depending on the acid used, various changes in the nature of the surface can be obtained, e.g., after using strong sulfuric acid, the porosity and surface area increase significantly, and the halloysite modified in this way has great potential for use as a catalyst. In turn, weak acids, such as acetic acid or acrylic acid, act primarily on the inner aluminum-hydroxide layer, and the halloysite modified in this way is characterized by an increased size of the open surface. The research shows that the inner diameter of the nanotube increases from 13.8 nm to 18.4 nm, thus increasing the load capacity of the halloysite by 77.8% [18]. In turn, modification with the use of alkalis has a stronger effect on the outer layer of silica [19], and the modified halloysite was studied for, among other purposes, for the adsorption and desorption of ofloxacin (drug active against gram-negative bacteria). In contrast to the acid modification, a decrease in drug release was observed with increasing base concentration [19].

In turn, active substances in the form of organic compounds can bond to the surface of halloysite by the reaction of complex formation, an attraction resulting from weak hydrogen interactions between hydrogen atoms of various functional groups and OH^−^ groups on the surface of halloysite, but it can also be an electrostatic attraction, when there are uncompensated electric charges on the molecule, or finally, the geometric “entrapment” of the molecule inside the nanotube.

There are numerous reports in the literature in which halloysite was modified with such organic compounds as: poly (*N*-isopropyl acrylamide [20], chitosan [21], triazole [22], cysteamine [23], cyclodextrin [24,25] and doxorubicin [26]. There are also reports describing the possibility of using curcumin (CUR) [27], which has antioxidant and anti-carcinogenic properties [28,29,30]. Many papers describe the curcumin “loading” process and the use of such a combination as a prodrug, i.e., an inactive substance that is activated in the body’s environment, which allows, among other things, an increase in the permeability of the drug through the cell membrane, a change in the solubility of the drug, and prolongation of the active substance efficiency [21,22,23,24,25,31]. There are also reports describing the physical and chemical properties of curcumin-modified halloysite [32], but no papers have been found reporting the incorporation of gentamicin (antibiotic) into the carrier modified in such way. It was reported, however, that curcumin binds to halloysite nanotubes as a result of the formation of the HAL-CUR complex, which leads to an increase in surface roughness, but which also locates inside the nanotube lumen. Such a drug delivery system is highly stable, and only 1% of curcumin has been found to be released after 24 h [32]. A completely different approach to the organic modification has been undertaken in recent years using organic polyamidoamine dendrimers [33], and the drug delivery system prepared in this way showed a better ability to bind the active substance and its prolonged release compared to the neat but silanized halloysite [33].

Nevertheless, it has been shown that the modification of halloysite with inorganic compounds can also be effective and increase the loading capacity, as noted in the case of ibuprofen incorporated into halloysite modified with APTES ((3-aminopropyl) triethoxysilane) and silane. Such a modification had a very positive effect on the release kinetics of this active substance, and its release time was 2.5 times longer and amounted to 50 h compared to unmodified halloysite, whose release time was only 20 h [34,35]. Other studies have shown that halloysite bi-functionalized with two ligands—folic acid and fluorochrome—and doped with the anti-cancer drug methotrexate, showed a high efficiency in drug loading and a prolonged release time [36]. Other research has also revealed that poly (*N*-isopropylacrylamide) grafted onto halloysite for thermo-responsive curcumin release resulted in targeted release of the active species into the intestine [20].

Another interesting research problem related to the modification of halloysite nanotubes is microstructural modification, which creates, for example, the possibility of delamination, leading to expansion of the nanotubes, and thus to increasing the specific surface area and adsorption of active substances. A similar strategy has been successfully tested for carbon nanotubes [37], and this work assumes the possibility of mapping similar phenomena for halloysite tubes. Hence, this line of research seems to be new, and our work is most likely the first such approach to surface modification of halloysite, as no similar work was found in the literature review.

The aim of the work was to examine the possibility of using modified halloysite nanotubes as a gentamicin carrier and to determine the usefulness of the modification in terms of the effect on the amount of the drug attached, its release time, but also on the biocidal properties of the carriers. The work includes a comprehensive comparison of various methods of halloysite surface modification, including acidic, alkaline, organic and nanotube delamination. The work also attempts to understand how the substances used for modification interact with halloysite and whether the modification changes the properties and morphology of nanotubes or if that process affects the desirability of the location of the active substance in such a carrier. The obtained materials were tested to determine the effects of surface modification and of the materials’ interaction with the introduced antibiotic on the biological activity of the carrier, the kinetics of drug release, and on the antibacterial activity against *Escherichia coli* Gram-negative bacteria (reference strain). For all materials, structural changes were examined using infrared spectroscopy (FTIR), X-ray diffraction (XRD), as well as thermal differential scanning calorimetry with thermogravimetric analysis (DSC/TG) was also performed. The samples were also observed by transmission electron microscopy (TEM) for morphological changes after modification and drug activation.

## 2. Materials and Methods

Dunino halloysite (HDU) extracted from the Polish mine in Dunino near Legnica (Poland), was used as raw material, after a purification process carried out by PTH Intermark, Gliwice (Poland). Unmodified Dunino halloysite was the reference material, also constituting the base for obtaining materials modified and intercalated with the drug. The following reagents were used to modify the surface of halloysite nanotubes: hydrogen peroxide approx. 30% p.a. by POCH Gliwice (Poland), potassium persulfate p.a. by POCH Gliwice (Poland), sodium hydroxide p.a. by STANLAB—Chemical Center, Lublin (Poland), sulfuric acid 98% p.a. by CHEMPUR, Piekary Śląskie (Poland), orthophosphoric acid 85% p. by CHEMPUR, Piekary Śląskie (Poland), and curcumin powder (from *Curcuma longa*, ≥65% (HPLC)) by Sigma-Aldrich (Poland). After the modification of the halloysite surface, the materials thus obtained were subjected to the intercalation process with the aminoglycoside antibiotic, i.e., gentamicin, in the form of a water-soluble sulphate powder from Pharma Cosmetic, Kraków (Poland).

### 2.1. Modification of Halloysite Nanotubes

The alkaline modification of halloysite (H5) was carried out in a process in which 5 g of neat halloysite powder was introduced into a 4 molar NaOH solution and subjected to ultrasonic mixing. This mixture was placed under vacuum for 1 h, and then washed three times with distilled water by sedimentation. The last step was the drying of the powder in a laboratory dryer at 50 °C. The acidic modification of halloysite (H1) was carried out according to the same scheme, using a 4-molar solution of H_2_SO_4_. In turn, the modification by oxidative expansion with sulfuric and phosphoric acids (H2) was carried out in two stages, with 10 g of halloysite prepared in the first phase and stirred into 100 mL of H_2_O_2_ with an ultrasonic bath for 60 min, and then in the second phase 30 mL of sulfuric acid was slowly added while stirring. The suspension was then washed with deionized water and dried for 12 h at 65 °C. To prepare the modification (H3), which was the second stage of acid modification, 5 g of the previously prepared powder were taken and mixed in 50 mL of H_2_O_2_ for 60 min with the use of an ultrasonic cleaner. Then, 15 mL of orthophosphoric acid was added and stirring continued for 60 min to elute the sulfuric acid from the solution, while stirring. The slurry was then washed with deionized water and dried. The modification with curcumin (H6), due to the low solubility of curcumin in water, was carried out in an ethanol environment, and 5 g of halloysite and 1 g of curcumin were added to 200 mL of ethanol. The mixture was stirred for 24 h at 50 °C, then washed with distilled water and dried in a laboratory oven at 50 °C. Ammonium persulfate modified halloysite (H4) was obtained by preparing a suspension of 75 g of ammonium persulfate in 45 mL of ultrasonically stirred sulfuric acid, to which 15 g of halloysite was then added and mixed again. The mixture was left to swell for several hours and then dried in a laboratory drier. After drying, all materials were pulverized in an agate mortar.

### 2.2. Preparation of Modified Halloysite-Drug Conjugates

For every 5 g of modified halloysite powder, 0.625 g of gentamicin was added according to the CEC of the starting halloysite (11.0 ± 0.6 mol/100 g). The powders were mixed with gentamicin in 100 mL of distilled water and then stirred at 50 °C for 24 h. After mixing, the powders were dried in a laboratory drier at the temperature of 50 °C, and after drying, they were ground in an agate mortar. All combinations of modified halloysite with gentamicin (H1G–H6G) and a reference sample consisting of a combination of neat halloysite and gentamicin (H0G) were prepared in this way. Table 1 contains a summary of the abbreviations and detailed descriptions of individual samples prepared for testing (starting powders, powders after modification and intercalation).

### 2.3. Infrared Spectroscopy (FTIR)

Infrared spectroscopy was used to determine the presence of characteristic functional groups that could provide information on how the drug bonds to the halloysite surface. Pellets containing 2 mg of the test powder dispersed in 400 mg of KBr, formed in a hand press under a pressure of 9 MPa, were made for measurements. The pellets made in this way were examined in the Vertex 70V Bruker spectroscope in the measuring range of 4000–400 cm^−1^.

### 2.4. X-ray Diffraction (XRD)

The XRD method was used to observe structural changes occurring in halloysite powders subjected to various modifications, as well as changes in the size of the crystallites in the powder. Measurements were made using a PANalytical Empyrean X-ray diffractometer.

### 2.5. Thermal Analysis (DSC/TG)

Differential Scanning Calorimetry (DSC) was used to determine thermal properties, while thermogravimetric analysis (TG) was carried out to investigate mass loss with increasing temperature. The measurements were conducted in parallel in the STA 449 F3 Jupiter thermal analyzer by Netzsch, in the measurement temperature range from 20 to 420 °C, at the temperature increase rate of 10 °C/min.

### 2.6. Transmission Electron Microscopy (TEM)

The TEM was used to observe the morphology of the powders and possible changes that may have occurred after drug modification and intercalation of halloysite nanotubes. The measuring instrument was a JEOL JEM-1011 Transmission Electron Microscope (TEM) from JEOL Ltd. Japan, and the samples were coated with carbon prior to the experiment.

### 2.7. Drug Release

The in vitro gentamicin release tests from a halloysite matrix were performed according to the United States Pharmacopeia: Content of gentamicin sulfate [38]. The nanotubes and modified nanotubes intercalated with gentamicin were incubated in a phosphate buffer (PBS), pH 7.4, at 37 °C for 20 days, maintaining a ratio of 1 to 5 × 10^−6^ of buffer volume to powder mass. At each measurement, 1 mL of supernatant was taken from the suspension and the drug content was determined spectrophotometrically (FLUOstar Omega, BMG LABTECH, Ortenberg, Germany). The release of gentamicin was evaluated based on the intensity of the maximum absorbance of the PBS fluid, measured at λ = 330 nm, and on its correlation with the prepared calibration curve for the drug. Two specific chemicals, namely o-Phthalaldehyde and β-mercaptoethanol (both from Sigma-Aldrich, St. Louis, MO, USA), were used to determine the gentamicin released during in vitro incubation, and the results represent the average of three measurements. To the collected supernatant (200 µL) a mixture of 2-propanol (220 µL) (POCH, Gliwice, Poland) and OPA solution (80 µL) was added. The whole was mixed and incubated for a minute at room temperature, and after this time the samples were diluted by adding 500 µL of a 50% solution of 2-propanol with methanol (POCH, Gliwice, Poland). The concentration of the drug released into the PBS solution was determined using the Beer-Lambert law, and the calibration curve was prepared based on solutions of known concentrations of gentamicin (0 ÷ 1000 μg/mL). According to this procedure, the amount of the drug released (%) was presented as a function of the release time (days). Based on the data available in the drug database [39], which describes the maximum daily dose of the drug (10 ÷ 12 μg/mL), the obtained gentamicin concentration in the PBS solution was recalculated to the maximum value of the drug (10 μg/mL).

The loading efficiency (*LE*) of halloysite was calculated according to Formula (1) [40]:*LE* (%) = (*M*_t_/*M*_0_)∙100(1)
where: *M*_t_—mass of the drug, *M*_0_—total mass of the powder sample after drug loading.

The kinetic of drug release was evaluated based on the zero order, Korsmeyer-Peppas, modified Korsmeyer-Peppas and Higuchi models, according to the Equations (2)–(5):

Zero order model [41,42]:*f*_t_ = *f*_0_ + *K*_0_·*t*(2)

Korsmeyer-Peppas model [1,43]:*f*_t_ = *K*_k_·*t^n^*(3)
modified Korsmeyer-Peppas model [44,45]:*f*_t_ = *K*_m_·*t^n^* + *C*(4)

Higuchi model [46,47]:*f*_t_ = *K*_h_·*t*^0.5^(5)
where *f*_t_ is the amount of drug released in time *t*, *f*_0_ is the initial amount of drug in the solution, *K*_0_, *K*_k_, *K*_m_ and *K*_h_ are the zero order, first order and Higuchi release constants respectively, expressed in units of concentration/time, *t* is the time of release, *n* is the release exponent, characteristic of the release mechanism, and *C* represents the burst effect in the release.

### 2.8. Antibacterial Testing

For the antibacterial tests, pellets with a diameter of 6 mm, a height of 1.5 mm and a weight of 0.1 g each were made from the obtained powders. The biocidal activity of the obtained conjugates and reference samples was tested using the disk diffusion method against Gram-negative bacteria *Escherichia coli*. Bacterial cultivation was carried out for 24 h on Oxoid’s Columbia Blood Agar in Petri dishes at a temperature of 35 ± 2 °C, in an atmosphere of 5% CO_2_. Three to five colonies of bacteria were suspended in a physiological NaCl solution (0.9%) until a suspension with a density of 0.5 McFarland was obtained, which corresponds to approximately 1.5 × 10^8^ CFU. Then, using a sterile swab, the suspension was inoculated onto Oxoid’s Muller Hinton Agar in 100 mm diameter dishes. After drying, the plates were covered with sterile tissue pads, and on them were placed pellets made of the tested powders. The diameter of the pellets and tissue pads was 6.0 mm. An antibiotic-free comparative blank disc and a blotter disc soaked in gentamicin sulfate solution (acc. to EUCAST 10 μg gentamicin solution [48]) were also prepared and subjected to the same test. The dishes with the discs were placed in the incubation chamber at 35 ± 2 °C under aerobic conditions. After 20 h, the diameters of the zones of inhibition of bacterial growth around the pellets were measured with a caliper. The test for each sample was repeated three times, and the measurement results were the average of three replicates.

## 3. Results and Discussions

### 3.1. Infrared Spectroscopy (FTIR)

The FTIR spectra of all tested samples are summarized in Figure 1 and Figure 2, while Figure 1 shows the results of FTIR measurements for the modified halloysite (H1–H6 samples), and Figure 2 shows the spectra of the modified halloysite-drug conjugates obtained with gentamicin (samples H0G–H6G). The figures also include the FTIR spectrum registered for neat halloysite (H0 in Figure 1) and for the conjugate of neat halloysite with the drug (H0G in Figure 2).

The infrared spectrum of neat H0 halloysite, shows a characteristic absorption band with a maximum at 470 cm^−1^, indicating the presence of Si-O-Si bending vibrations. The absorption bands with the maximum at the wavenumber of 541 cm^−1^ correspond to the Al-O-Si vibrations, while the bands with the maxima at the wavenumbers 804, 1033, and 1118 cm^−1^ indicate the presence of Si-O bonds. The effect visible at 916 cm^−1^ indicates the presence of hydroxyl groups (Al-OH), and the bands between 900 and 1200 cm^−1^ are caused by vibrations of the Si-O-Al aluminum silica bridges or Si-O-Si silicon oxygen bridges. A faintly visible band of extremely low absorbance with a maximum at 1668 cm^−1^ indicates the presence of strongly bound structured water, which confirms the presence of partially hydrated halloysite in the samples. Bands with maxima at 3624 and 3699 cm^−1^ indicate the presence of interactions between OH groups connecting two aluminum atoms, which is characteristic of halloysite and corresponds to the literature data describing the bonds in this material [1,4,49,50,51,52].

The modification of halloysite with sulfuric acid (H1) does not substantially affect the shape of the spectrum, which remains similar to that of the neat raw material. No additional bands are observed, and the difference only appears in the lower absorbance values over the entire measuring range. The band with a maximum at 1668 cm^−1^ for neat halloysite, indicating the presence of structured water, shifted and decreased, which may indicate the transformation of hydrated halloysite into metahalloysite. The next three modifications (H2–H4 samples), aimed at expanding halloysite, are characterized by a clear reduction, or almost disappearance of the bands at 3624 and 3699 cm^−1^, which proves a strong reaction of the modifiers with OH groups in the halloysite. The bands responsible for the vibrations of the Si-O-Al silica bridges or Si-O-Si silicon oxygen bridges are also significantly reduced, which may be related to the elution of silicon and aluminum atoms from the mineral.

In the spectrum obtained for halloysite modified with sodium hydroxide (H5), there are visible bands at 3624 and 3699 cm^−1^ wavenumbers, which are responsible for the presence of OH groups. As expected, the alkaline should wash out mainly silicon [19], so the bands responsible for vibrations of connections involving silicon with the peaks at 470, 804, 1033, and 1118 cm^−1^ have much lower absorbance.

In the H6 sample, modified with curcumin, additional bands of low absorbance appeared at wavenumbers 1232, 1263, 1307, 1450, 1537, and 1631 cm^−1^, which is related to the presence of an additional organic compound. The vibrations represented by the band with the maximum at the wavenumber of 1631 cm^−1^ are related to the vibrations of the C=C double bonds. The vibrations characteristic of curcumin at the wave numbers of 1023 cm^−1^ (related to C-C vibrations) and 1120 cm^−1^ (vibrations of bonds in OH groups) most likely overlap with the vibrations coming from halloysite in the same range and their separation is not possible. The band appearing at the wavenumber of 1450 cm^−1^ indicates the presence of C-H bonds, and neither vibrations at the wave number 2918 cm^−1^ (C-H stretching vibrations) nor 2954 cm^−1^ (vibrations of the COOH group) were observed. It should be noted, however, that even on the spectrum of pure curcumin, these bands are characterized by a very low absorbance [53]. There were also no new bands resulting from Si-C or Al-C bonds, which proves that curcumin did not form chemical bonds with the halloysite surface, but rather adsorbed on its surface.

Figure 2 shows the results of infrared spectroscopy studies for the prepared halloysite-drug conjugates, and the problem with interpreting these results is that the characteristic effects associated with vibration of the chemical bonds in gentamicin are in the same ranges as the vibration of the bonds in the halloysite. The spectrum of gentamicin sulfate shows absorption bands in the 900–1300 cm^−1^, range, which correspond to the vibration of the C-N and C-O bonds. The absorption bands visible at the wavenumbers 1400–1700 cm^−1^ are related to the vibration of the N-H, C-H and C-N bonds, while those in the range 2800–3000 cm^−1^ correspond to the vibration of the C-H bonds. The bands in the range 3100–3500 cm^−1^ indicate the presence of amino groups and are related to N-H stretching vibrations [54,55,56]. The sample H0G, containing neat halloysite and gentamicin, shows only a change in the range 1100–1140 cm^−1^, and the sharp absorption band was flattened by overlapping bands from C-N and C-O bonds in gentamicin. In the H1G sample (modified with sulfuric acid), changes at 1660 cm^−1^ can be observed, which indicate N-H, C-H and C-N vibrations from gentamicin sulphate. There is a similar flattening effect of the sharp band in the range of 1100–1140 cm^−1^ as in the H0G sample. In the H2G sample (expanded with sulfuric acid in the presence of perhydrol), the vibrations related to the presence of Si disappear (modification effect), and in the range 990–1200 cm^−1^, a very similar shape of the spectrum can be clearly observed as in the pure gentamicin sample. or the H3G sample; expanded with orthophosphoric acid, there is a clear increase in absorbance in the broad range of 2500–3500 cm^−1^, as well as in the wide band characteristic for gentamicin with a maximum at the wavenumber 1670 cm^−1^. There is the same flattening effect of the sharp band in the 1100–1140 cm^−1^ range as in the previous samples. The H4G sample, modified with ammonium persulfate, shows the presence of an additional band corresponding to the wavenumber of 1479 cm^−1^, which is neither related to the presence of the antibiotic, nor is present in the spectrum before the addition of gentamicin. This may be due to the presence of impurities or the formation of a completely new bond between the modified sample surface and gentamicin sulfate. A strong increase in absorbance in the broad range 2500–3500 cm^−1^, related to the presence of the antibiotic in the sample, is also noticeable. No effects from gentamicin are observed in the sample modified with NaOH (H5G); however, it should be noted that they may overlap with the vibration bands in the halloysite and thus be invisible, or the antibiotic has not bound to the surface and has been completely degraded in the alkaline environment. The sample modified with curcumin (H6G) abounds in absorption bands in the range of 400–2000 cm^−1^, and in addition to the previously observed bands with maxima for the wavenumbers 1232, 1263, 1307, 1450, 1537, and 1631 cm^−1^, there is another band with the maximum at the wavenumber of 1650 cm^−1^. It may come from gentamicin, but in this case the band is sharp, while in pure antibiotic it is wide and rounded. It may also indicate the formation of a new chemical bond between curcumin and gentamicin sulfate.

In most samples, no new bonds involving silicon or aluminum were observed, which may indicate that the antibiotic did not form new chemical bonds on the halloysite surface. It is also possible that new bonds were formed between curcumin and gentamicin, as well as in the ammonium persulfate expanded halloysite sample.

### 3.2. X-ray Diffraction (XRD)

Figure 3 shows a summary of the X-ray diffraction patterns for the neat H0 halloysite sample and the modified halloysite samples (H1–H6). The low angular range in which this test was carried out (2–15° of the 2θ angle) results from the characteristics of clay minerals in which the interplanar distances are large, so the analytical peaks occur at low values of the 2θ angle. From the X-ray diffraction pattern made for the H0 sample of neat halloysite, it can be concluded that it consists mainly of non-hydrated halloysite with an interplanar distance d_001_ = 7 Å (at about 12° 2θ), with crystallite dimensions from 7.8 to 18.7 nm [57].

In the case of the H1 sample modified with sulfuric acid, some metahalloysite is still observed, containing crystallites of 16.1 nm and 8.3 nm. The sample also features larger crystallites with a greater interplanar distance d_001_ = 13.3 Å, and these may be etched tubular nanotubes, swollen as a result of the weakening of the attractive forces of individual layers as a result of aluminum leaching from the octahedral inner layer.

The X-ray diffraction pattern of the H2 sample, expanded with sulfuric acid in the presence of hydrogen peroxide, shows that in addition to the basic metahalloysite nanotubes, new, exceptionally large crystallites with a diameter of 133 nm and an interplanar distance close to 7 Å have appeared. As the interplanar distance has not changed, this may indicate that the nanotube unfolded without swelling of the layered structure. The amorphous nature of this sample is also increased, as evidenced by lower diffraction peaks than in the previous cases.

The X-ray diffraction pattern of sample H3, subjected to double expansion with sulfuric and phosphoric acid, shows the appearance of a new, intense peak for the angle of 2θ = 6.5°, which indicates a change in the crystal structure of the tested sample. The interplanar distance corresponding to this peak is 13.4 Å, which may suggest that the halloysite structure swelled and the forces between the tetrahedral and octahedral layers were weakened. Expansion with ammonium persulfate of sample H4 completely destroyed the crystal structure of halloysite. It can be assumed that in the range of very low angles there is a diffraction peak, but the assessment is very uncertain. It could indicate an intense swelling and an increase in the interplanar distance compared to metahalloysite.

Modification with sodium base of sample H5 did not significantly affect its crystal structure, and its X-ray diffraction pattern in the range of 10–15° of 2θ angle is very similar to the diffractogram obtained for neat H0 halloysite, with almost identical crystallite dimensions and interplanar distances. The H5 sample is dominated by metahalloysite with crystallites of 19 nm and 8.3 nm. There is a faint peak indicating the presence of a phase with an interplanar distance of about 14 Å, which may be represented by etched halloysite, in which the base has removed silicon from the tetrahedral layer, increasing the interplanar distances. In turn, the sample modified with curcumin H6 additionally shows a double peak in the X-ray diffraction pattern around the angle 2θ = 14.6°, representing crystallites with an interplanar distance of d_001_ = 6 Å.

Figure 4 shows X-ray diffraction patterns of neat halloysite with gentamicin (H0G) and modified halloysites intercalated with gentamicin (H1G–H6G). The X-ray diffraction pattern of neat halloysite after gentamicin intercalation (H0G) indicates that its microstructure did not change significantly, and the crystallite size and interplanar spacing remained almost unchanged. On the other hand, in the H1G sample, after the addition of the antibiotic, the powder became more amorphous, and the diffraction peak was much lower and wider than that of the modified H1 sample. Crystallite sizes also increased by 5 and 2 nm respectively, although the interplanar distances remained the same. The X-ray diffraction pattern of the H2G powder, which is a combination of halloysite expanded with sulfuric acid in the presence of hydrogen peroxide with gentamicin, contains a single diffraction peak for the same 2θ, for which a double peak previously occurred (the second one was related to the presence of crystallites 133 nm and d_001_ = 6.99 Å). This peak disappeared upon addition of gentamicin, but a new peak appeared for the lower 2θ angle, indicating the presence of crystallites at the same 133 nm size, while increasing the interplanar distance (13.29 Å). After adding gentamicin to the sample expanded with sulfuric and phosphoric acid (H3G), the dimensions of the crystallites assigned to the peak with a maximum at 2θ = 12.8° increased from 21.6 to 133 nm, while those related to the peak with a maximum at 2θ = 12.2° decreased from 24.2 to 14 nm, and the interplanar distances remained almost unchanged. The results obtained for the sample H4G are very surprising, and it was difficult to find any peak on the diffraction pattern before the addition of gentamicin, which indicated that the structure of the sample was damaged and was highly amorphous. However, after the addition of gentamicin, strong and sharp peaks can be observed, indicating an ordered structure, which may indicate that after long mixing with the antibiotic at 60 °C the modifier recrystallized. The addition of gentamicin to the sample modified with sodium hydroxide (H5G) did not cause significant changes, and all peaks before and after drug intercalation are in the same positions and have similar intensities. The differences in the size of crystallites are of the order of a few nanometers, while the interplanar distances have not changed. The curcumin-modified sample after the addition of gentamicin (H6G) also underwent only slight changes, and the diffraction patterns before and after drug intercalation are similar. The differences in the size of the crystallites are very small, and only for the peak corresponding to the angle 2θ = 12.1° the crystallites are larger by 6 nm, while the interplanar distances remained unchanged.

### 3.3. Thermal Analysis (DSC/TG)

Figure 5 shows the results of the thermogravimetric analysis (TG) performed for neat halloysite (H0) and halloysite samples subjected to various modifications (H1–H6).

The graph shows that in the samples H0 (neat halloysite), H1 (halloysite modified with sulfuric acid), and H5 (halloysite modified with NaOH) changes related to weight loss proceed in a similar way. Between room temperature and 100 °C, only a loss of mass related to the evaporation of free water can be observed, then up to 400 °C a very slight loss can be seen, while above this temperature the process of evaporation of water bound in the structure of the mineral begins. For neat halloysite, the weight loss was as high as 17.29%, for the H1 sample—19.16%, and for the H5 sample—18.19%. A completely different course of the TG curve can be observed in the case of halloysite subjected to expansion with sulfuric acid in the presence of hydrogen peroxide (H2). In this case, there is a continuous weight loss in the temperature range 20 to 800 °C, the most intense to a temperature of 122.5 °C, and the total weight loss of this sample was as high as 42%. In the other sample, subjected to additional modification with phosphoric acid (H3), there is an intense decrease in mass to approx. 120 °C, related to the evaporation of water, and changes related to the removal of water bound in the crystal structure can be observed at a temperature of approx. 490 °C. There is much lower content of structured water than in neat halloysite, which means that the sample consisted mainly of metahalloysite. The process of structural water loss is also shifted towards higher temperatures by about 20 °C, and the process is completed at 550 °C, which indicates an increase in the stability of the mineral. In the case of the H6 powder, modified with curcumin, a large decrease in the mass of the sample in the temperature range 280–520 °C was observed, representing 29.55% of the mass of the sample, which is related to the decomposition of curcumin to 400 °C, followed by the loss of structured water.

Figure 6 shows the results of tests carried out with the use of Differential Scanning Calorimetry (DSC) in the temperature range 20 to 800 °C. The weight losses that were recorded in the TG study translate into effects caused by changes in the heat flux related to the endo- or exothermic process leading to the mass changes recorded on the DSC curves. DSC analysis also allows for a more accurate determination of the chemical and physical processes that are not related to weight loss and are not visible on the TG curve. Thanks to this, it is possible to determine what type of process takes place with a given weight loss.

The DSC curve for neat H0 halloysite shows an endothermic peak at 494.1 °C, which is related to the release of structured water. Later, the course is constant, and only this one thermal effect is observed in the entire studied temperature range. The DSC curve for the sample H1 (modified with sulfuric acid) and H5 (modified with NaOH) is similar, and the endothermic peaks associated with the process of releasing water from the structure of the mineral appear at the same temperature point. In the case of the H2 sample (expanded with sulfuric acid in the presence of hydrogen peroxide), a completely different course of the curve can be observed. At the very beginning, there are two endothermic peaks recorded for the temperatures 108.7 °C and 122.5 °C. The first one is related to the evaporation of free, unbound water, while the second one may be responsible for the decomposition of sulfuric acid, which, according to the literature data, begins to decompose at about 130 °C [58] with the release of water with sulfur oxide. In the case of this sample, the endothermic effect responsible for the release of water from the halloysite crystal structure is almost imperceptible. This peak is followed by a decrease in the course of the DSC curve combined with a weight loss to the end of the measuring range, with no visible bends. Similar effects, but for lower temperatures, i.e., 83.7 °C and 120.6 °C, occur for halloysite subjected to double acid modification (H3). These two endothermic effects are related to the evaporation of water and the decomposition of phosphoric acid into water and phosphorus oxide. There is also an endothermic peak at 493.9 °C corresponding to the release of structural water. The DSC curve descends to the end of the measuring range, which means that the powder continues to oxidize with the release of gaseous products. In the H4 sample subjected to expansion with ammonium persulfate, apart from the endothermic effect around 60 °C related to evaporation of moisture and a peak of approx. 100 °C, which may be related to the evaporation of water adsorbed with the decomposition of ammonium persulfate, there is also an exothermic peak at a temperature of approx. 413 °C, which can come from the burning of perhydrol or impurities in the sample. Sample H6 modified with curcumin shows the most complicated course of the DSC curve, and at a temperature of about 328 °C, curcumin burning begins, manifesting itself as a high exothermic peak. The second exothermic effect appears at the temperature 364 °C, and when this temperature is exceeded, the curcumin burning process is completed [59]. At the temperature 505 °C, structural water is released (endothermic effect), but it is worth noting that this process takes place at a higher temperature than in the case of other samples, which proves the increase in the stability of the mineral.

Figure 7 and Figure 8 show the TG and DSC curves for pure gentamicin, a gentamicin intercalated reference sample, and halloysite samples modified prior to gentamicin intercalation, respectively.

The curves for the H0G, H1G, and H5G samples followed a comparable course, similar to the course of the TG curves before the drug intercalation, and the measured weight loss of the three samples ranged from 20–26%. The weight loss is about 10% greater than in the samples without the antibiotic, and the already described effects of water evaporation, modifying substance decomposition, structured water release, and gentamicin burning above 300 °C overlap in the tested case. In the H2G sample (expanded with sulfuric acid in the presence of perhydrol), a continuous weight loss of up to 48% was observed in the entire temperature range tested. It is an expanded sample, the structure and individual layers of which have been destroyed and delaminated, respectively, and such a large loss in weight may indicate a high content of adsorbed antibiotic. The H3G sample shows the greatest weight loss up to a temperature of 120 °C, and the curve course is very similar to the curve recorded for the samples before the addition of the antibiotic. In this TG curve there is no clear reflection of the bound water release process in the halloysite crystal structure, and the weight loss is 28%. The H4G sample shows the greatest weight loss at the level of 70%, which indicates that with increasing temperature, many volatile components were released. These are ozone, ammonia, nitrogen, and sulfur oxides from the decomposition of ammonium persulfate [60] and carbon dioxide associated with the melting and decomposition of gentamicin.

For the H6G sample, modified with curcumin, there is an intense weight loss in the temperature range of 220–500 °C associated with curcumin burning, which is also combined with the loss of weight associated with the release of structural water at a temperature above 400 °C, and the total weight loss of this sample is 46%. The DSC curve for sample H0G (reference sample containing neat halloysite with gentamicin), compared to sample H0 (neat halloysite), shows an overlapping of two thermal effects—exothermic from the burning of gentamicin and endothermic associated with the evaporation of water trapped in the crystal structure. Burning of gentamicin begins at around 200 °C, which is visible in all samples except H6G. The H1G sample follows the DSC curve of the H0G sample, but the endothermic effect is less visible, and it is possible that its composition is dominated by metahalloysite. In the H2G sample (a conjugate of ammonium persulfate-modified powder with the drug), a relatively low endothermic peak attributed to the evaporation of adsorbed water at the temperature of about 100 °C is visible. The DSC curve for the H2G sample also shows an exothermic peak at a temperature of about 480–500 °C, which may be caused by the burning of hydrogen peroxide, gentamicin, or sulfuric acid, although these compounds alone have lower decomposition temperatures. The H3G sample (double-expanded with sulfuric and phosphoric acid in the presence of perhydrol) has a slight endothermic effect at 120 °C, most likely related to the decomposition of the perhydrol used for the modification. The endothermic peak related to the release of structured water is small, which may indicate a high metahalloysite content. It has also shifted by approx. 10 °C towards lower temperatures, which means that the thermal stability of the sample has decreased. The H4G sample (expanded with ammonium persulfate) shows a large endothermic peak related to the evaporation of adsorbed water at the temperature 100 °C. It may be related to the earlier decomposition of ammonium persulfate, which usually decomposes into ozone, ammonia, nitrogen oxides, and sulfur at a temperature of 120 °C [60]. This is also reflected in the fast and significant loss of sample mass in the TG plot. At the temperature of 220 °C, another, smaller endothermic effect appears, which may be associated with the decomposition of sulfuric acid with the release of sulfur oxide. The DSC curve for the H4G sample shows a small exothermic peak at a temperature of 460–480 °C, which may be attributed to the combustion of an organic substance originating from the modifier of halloysite nanotubes as well as gentamicin. In the H5G sample (a conjugate of NaOH-modified halloysite and gentamicin), as in the H0G sample, a thermal effect is observed, related to the burning of gentamicin at a temperature of 200 to 400 °C, followed by an endothermic peak indicating the release of structural water. The H6G sample is characterized by a strong exothermic reaction related to the burning of curcumin together with the antibiotic, reflected in the TG curve, and the endothermic peak derived from halloysite, which indicates the release of water from the structure of the mineral [61].

### 3.4. Observation of Changes in the Morphology of Powders after the Modifications

The studies using transmission electron microscopy allowed for the observation of changes in the morphology of the tested powders, as well as changes taking place on the surface of the halloysite nanotubes after their modification. Figure 9 presents microphotographs of samples of the unmodified H0 halloysite (starting powder) and modified halloysite (H1–H6) along with samples of unmodified and modified halloysite after gentamicin intercalation (H0G–H6G).

The TEM micrograph of the H0 sample (Dunino halloysite starting powder) confirms the fact that the Polish Dunino field abounds in grains of various morphological forms: tubes and flakes, as well as numerous flat-plate grains. Nanotubes are several hundred nanometers long and have a diameter of several dozen nanometers, and the flake grains range in size from several dozen to several nanometers.

In the case of sample H1 (modified with sulfuric acid), changes can already be observed on the surface, which is covered with large pits, which may indicate etching of the surface by sulfuric acid. As a result, halloysite expanded with sulfuric acid in the presence of hydrogen peroxide (H2) has a completely damaged surface, the nanotubes are broken, and amorphous clumps appear. The observed changes confirm the previously described results of structural studies carried out by means of X-ray diffraction. In the case of sample H3 (expanded with phosphoric acid), there was an increase in amorphism, as in the case of sample H4, expanded with ammonium persulfate. The grains of the H5 sample (NaOH modification) in terms of shape and size are like the reference sample H0, but there are traces of etching on the surfaces.

The modification with curcumin (sample H6) makes the grains “sticky”, which is visible in the microphotograph as a coating, and curcumin is present on the surface of the grains, taking the shape of a drop/bulge. No changes in the morphology of unmodified halloysite (H0G sample) were observed after drug intercalation. Similarly, in the H1G sample (modified with sulfuric acid + gentamicin intercalation), the grain morphology did not change; however, small clusters of spherical precipitates can be observed on flat grains, distributed unevenly on the surface, while no changes were observed inside the nanotube, which may indicate that the antibiotic did not penetrate inside. In the H2 sample (expanded with sulfuric acid in the presence of hydrogen peroxide with gentamicin), as before the drug intercalation, there is a largely amorphous clump of grains. On the microphotograph of the H3G sample (double acid modification in the presence of hydrogen peroxide intercalated with gentamicin), many aggregates of spherical particles, concentrated mainly at the edges of the grains, were observed. It is also possible that they filled the inside of the nanotubes or were present on the surface, perpendicular to the lumens. The largely amorphous sample of H4G, in which the modification with ammonium persulfate destroyed the crystal structure before the addition of the drug, was tightly surrounded by particles that held together individual grains. In the case of the H5G sample (modified with NaOH and intercalated with gentamicin), rectangular grains crystallized, which could constitute precipitates of sodium hydroxide. There are large differences in the morphology of this sample compared to the reference sample of neat halloysite intercalated with gentamicin H0G. The H6G sample, modified with curcumin and intercalated with the drug, showed large changes on the surface on which both curcumin and gentamicin had deposited, forming precipitation outside and possibly inside the nanotubes.

### 3.5. Drug Release and Kinetics

The amount of gentamicin loaded in the powders varies and strongly depends on the type of modification, as shown in Table 2; therefore, differences in drug release rates are clearly visible, as shown in Figure 10. It can be seen from the release curves that the exfoliated (recrystallized) form of HNT with gentamicin in the H4G sample shows the highest drug loading capacity. Ammonium persulfate acts as an expanding agent, thus destroying the HNT structure; ammonia, nitric oxide, and sulfur oxide are released during decomposition, but ozone free radicals may also appear. In addition, the environment during such a reaction is not acidified, which can lead to the deprotonation of Si-OH groups on the outer surface of HNT, which makes it negatively charged, and gentamicin sulfate can be the initiator of electrostatic interactions between damaged HNT surfaces. A large amount of gentamicin physically adsorbed to the surface of the halloysite flakes has a strong bactericidal effect and already is easily released, in the first hours of incubation.

In the case of HNT samples exfoliated with perhydrol (H2G and H3G), there is a similar release effect, however significantly lower than in the case of HNT exfoliated with ammonium persulfite. It is known from the literature [62], that at low pH, the positive charge of the inner lumen of the HNT is protonated, which leads to its expansion and facilitates the charging of the inner part of the HNT (hence the enlargement of the crystallites visible in XRD studies).

As a result, a larger amount of gentamicin remains in the center of the nanotube lumen, and a slightly smaller amount on the surface, but the degree of drug loading remains at a relatively high level (Table 2). This translates into a still high bactericidal effect, but also a lower release rate. The last group of materials is H0G, H1G, H5G, and H6G, which can in turn be characterized by a low loading efficiency, although in the case of H1G, H5G, and H6G, the modification process was performed. It seems that here only the HNT surface has been modified, which gives a bactericidal effect, but the release of the drug is at a very low level. Alkaline activation at room temperature (H5G) has no clear effect on the crystal structure of halloysite, and gentamicin sulfate is adsorbed on the negative surface of halloysite by electrostatic interaction—and by possible complexation of the drug with amorphous aluminum appearing in the lumen of halloysite nanotubes after NaOH treatment. Alkaline activation can extend the release time of the adsorbed drug, as is the case with the release of cationic drugs. In the case of preparation of conjugates using an organic linker, i.e., curcumin, in a neutral pH environment, there may be a competitive electrostatic interaction between the HNT surface and curcumin, which prevents the subsequent connection of gentamicin sulfate with HNT. Therefore, during thermal tests, large mass losses resulting from the combustion (decomposition) of an organic compound such as curcumin are observed. This interpretation is also supported by the low drug loading efficiency of the H6G powder.

The conclusions drawn above were supplemented by the analysis of the parameters of analytical models of gentamicin release kinetics calculated according to the four different models (Table 3), and the calculated parameters indicate that the *K* release constants generally take the highest values for the H2G, H3G, and H4G samples.

For these samples, the amount of drug released during the entire in vitro test was the highest and amounted to approximately 61.3%, 65.4%, and 90.1% for H2G, H3G, and H4G samples, respectively. It is worth mentioning that relatively high values of the kinetics constant K were obtained for the H5G sample, for which, however, the amount of released drug remained at a relatively low level of 35%.

To summarize, the release of gentamicin from all materials is a two- or even three-stage process, with different rates of drug release. For this reason, fitting with a zero-order model was only possible after separating the release plot into three regions. The first, most dynamic one, describes the immediate release (including the burst effect) in the initial 6 h, when the most loosely bound or readily available drug is released. After this time, the release stabilizes until, after about one day, the dynamics of the process accelerates again, which lasts until about the 10th day. This second acceleration of the release dynamics is related to the availability of the drug, which is more firmly bound to the halloysite structure inside the nanotube lumen. After this period, the process stabilizes again, and the maximum amount of drug delivered to the solution is reached. As is clear from the results, the modification of the halloysite surface has a significant effect on the amount of drug that can be released into the environment by rearranging the lumen structure of the nanotube and the availability of additional volume for the intercalated gentamicin. However, the modification of the halloysite does not substantially affect its ability to release the drug even further over time into the surrounding environment.

### 3.6. Antibacterial Tests

Table 4 presents the results of antibacterial tests expressed in the form of the diameter of the zone of inhibition of bacterial growth for neat halloysite and the samples of halloysite subjected to modifications (H0–H6), as well as for the samples intercalated with gentamicin (H0G–H6G). The diameter of the reference sample disc H0 was set at 7 mm, while the other discs had a diameter of 6 mm, and the result is the average of three measurements.

Figure 11 shows photos of the inhibition zones for individual materials after contact with *E. coli* bacteria, taken for the reference sample of pure halloysite H0 and samples of modified halloysite H1–H6 as well as for the samples of unmodified and modified halloysite after intercalation with gentamicin H0G–H6G. A close-up of sample H4 was also presented, which showed a very unusual zone of inhibition of bacterial growth.

From Table 4 and Figure 11 it is clear that the H3G and H5G samples showed the greatest inhibition zone for *E. coli* bacteria. Both samples are conjugates of halloysite modified with gentamicin, where in the sample H3G the halloysite was modified twice with sulfuric acid and orthophosphoric acid in the presence of hydrogen peroxide, and in the sample H5G with NaOH, respectively. It is worth noting that before the addition of the antibiotic, the H5 sample did not show a zone of inhibition of bacterial growth; therefore, the results obtained for the H5G sample are the most significant. For sample H3G, dissolution of the disc building material and its spilling around the pellet was observed, which consequently led to the formation of an inhibition zone equal to 15 mm even before drug intercalation. The H0G sample (neat Dunino halloysite intercalated with the drug) and the H1G sample (sulfuric acid modification with gentamicin) show exactly the same *E. coli* inhibition zone of 24 mm. Interestingly, the H1G sample has an additional surrounding from the dissolving pellet that was not formed before the addition of the antibiotic. Much better results were expected from this sample, as the modification with sulfuric acid alone led to a strong bactericidal effect. Therefore, it is assumed that this proves only that this type of modification may adversely affect the action of the drug. The dissolving effect of the pellet material, to a lesser extent, also occurs for H0G and H2G samples, which suggests that such pellets would not be stable in the living body environment. Modification of the H2G sample also negatively affected the action of gentamicin, and the measured diameter of the inhibition zone was larger, as expected, because the biocidal effect was already visible after the modification alone. The sample expanded with ammonium persulfate with the addition of gentamicin (H4G) showed no greater zone of inhibition than the reference sample, although the sample H4 without gentamicin showed bactericidal activity. It is also notable that the native H4 material showed a rather unusual (“strange”) zone of bacterial growth inhibition around the sample, in which a milky halo with microcolonies was visible, perhaps formed from migrating disc-building material, indicating partial growth inhibition. After intercalation with gentamicin, the H4G sample does not show a milky coating in the zone of inhibition, which appeared before the addition of the drug, but it should be remembered that sulfuric acid was used in the modification process, which may strongly affect the properties of gentamicin. Unfortunately, the H6G sample modified with organic curcumin and intercalated with the drug showed a smaller zone of inhibition than pure halloysite with gentamicin H0G.

The results of the antibacterial tests indicate that the best antibacterial activity is shown by the halloysite sample after modification with the sodium base and intercalation with the drug. It is important to note that the modification of H1G, H4G, and H6G samples could negatively affect the bactericidal properties of the antibiotic, while confirmation of the effectiveness of surface functionalization was obtained in the case of H3G and H5G samples. In the case of the H3G sample, a double acid modification was used, in which the sulfuric acid was washed out with weak phosphoric acid, thanks to which there was no inhibition of the antibiotic’s bactericidal properties for this sample.

## 4. Conclusions

The aim of the research was to evaluate the influence of halloysite surface modification on its properties and on the ability to intercalate with gentamicin. The analysis of the results allows for the formulation of the following conclusions:FTIR studies showed that in most cases no new bonds were formed between the modifiers and halloysite and between halloysite and gentamicin. Only in the samples H4G (modification with ammonium persulfate) and H6G (modification with curcumin) new absorption bands appeared, which may suggest the formation of new chemical bonds. The acidic and basic modification of the surface strongly interacts with OH^−^ groups and, as expected, washes out aluminum and silicon from halloysite.The XRD diffraction study showed that in all samples the majority phase is metahalloysite (dehydrated halloysite with an interplanar distance of 7 Å). Modification with ammonium persulfate (sample H4) completely destroyed the tubular structure of halloysite; it is even possible that the packages burst into single tetrahedral and aluminum octahedral silicon layers. Modification with curcumin H6 made the nanotubes twist more or led to filling their empty space.Thermal analysis DSC/TG showed that the modification of the surface affects the stability of the mineral, which is manifested by a change in the temperature of water release from the crystal structure. The reference sample (H0) and the samples modified with sulfuric acid (H1) and base (H5) showed similar behavior under the influence of temperature. The H6G sample is the only one that is not stable above 200 °C. The greatest weight loss occurred for the H4G sample, which means that it had the greatest amount of antibiotic adsorbed, and that may be due to the increased sample surface (breakdown of large grains into smaller ones).The TEM microscopy revealed a strong influence of the modification on the morphology of halloysite. After the modification with NaOH and the addition of gentamicin, the hydroxide precipitated, while the expansion with ammonium persulfate destroyed the tubular structure. Gentamicin appears in most cases as “drops” on the surface of the grains.Modification of the halloysite surface has a significant effect on the amount of gentamicin that can be intercalated and released into the surrounding environment; however, it does not significantly affect its ability to further delay drug release over time.Antimicrobial studies showed that the largest zones of inhibition of the growth of *E. coli* bacteria were observed for halloysite modified with sodium hydroxide (sample H5G). In turn, modifications with the use of sulfuric acid might have a negative effect on the biocidal activity of gentamicin, as the antibacterial activity noticed for H1G and H2G was not as high as expected. It has also been shown that leaching of sulfuric acid (in the example of double modification with sulfuric and phosphoric acid) reduces its negative impact on the antibiotic.

## Figures and Tables

**Figure 1 jfb-14-00167-f001:**
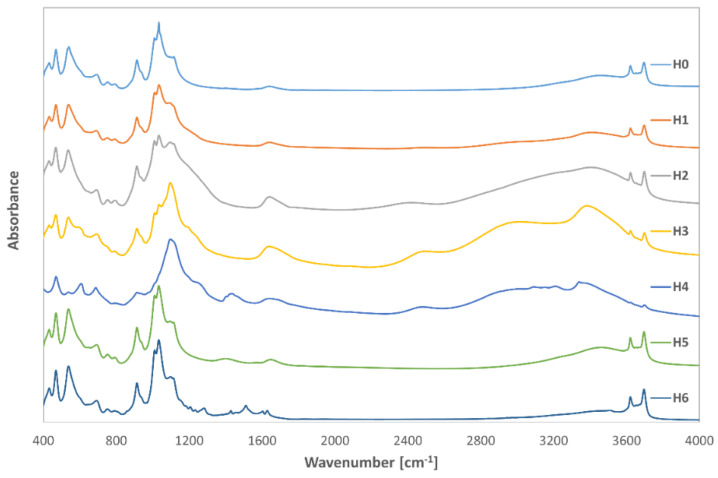
FTIR spectra of neat (H0) and modified halloysite (H1–H6).

**Figure 2 jfb-14-00167-f002:**
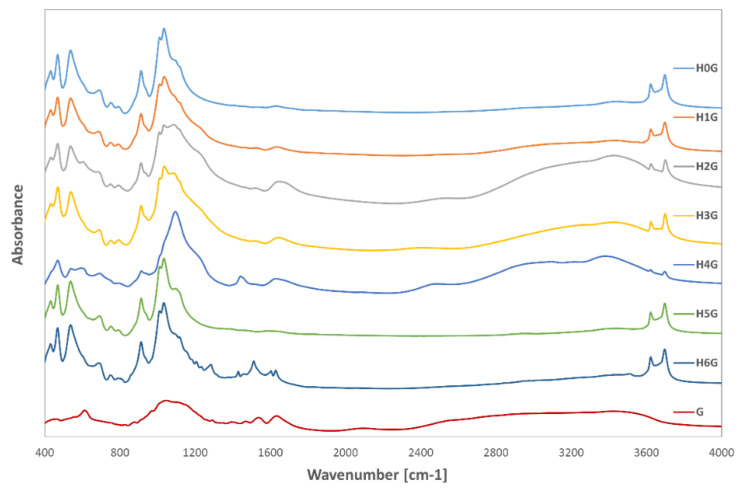
FTIR spectra for drug-halloysite conjugates (H0G), drug-containing modified halloysite (H1G–H6G) and pure gentamicin (G).

**Figure 3 jfb-14-00167-f003:**
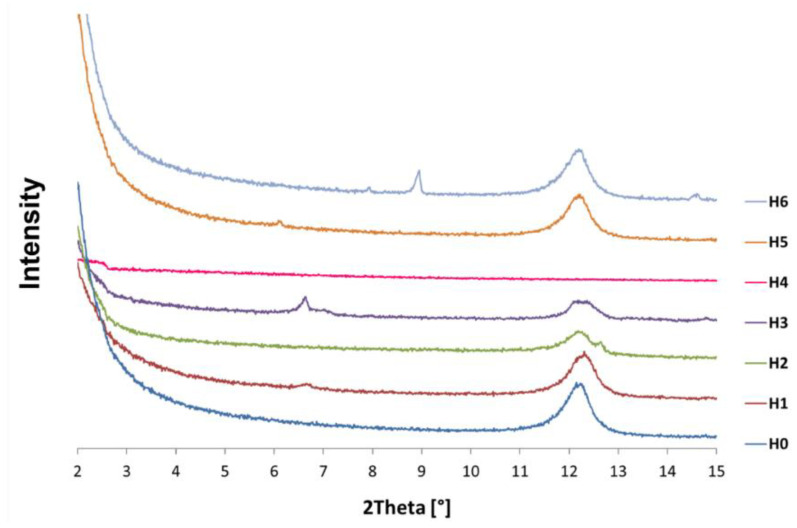
X-ray diffraction patterns of neat halloysite (H0) and modified halloysites (H1–H6).

**Figure 4 jfb-14-00167-f004:**
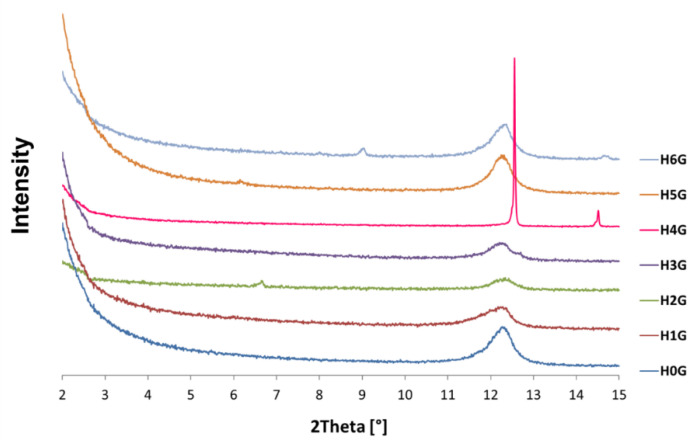
X-ray diffraction patterns of neat halloysite with gentamicin (H0G) and modified halloysites intercalated with gentamicin (H1G–H6G).

**Figure 5 jfb-14-00167-f005:**
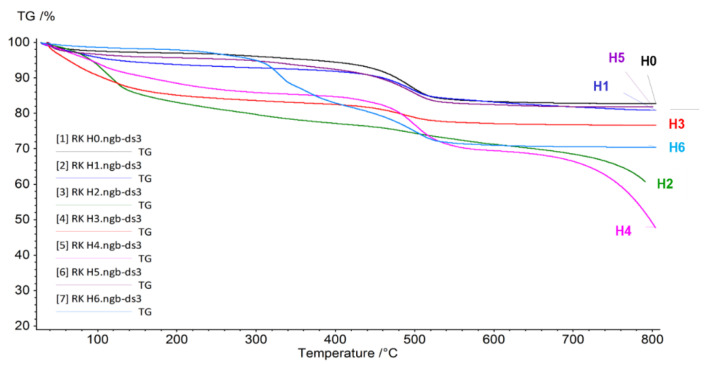
TG curves of neat halloysite (H0) and halloysite after modifications (H1–H6).

**Figure 6 jfb-14-00167-f006:**
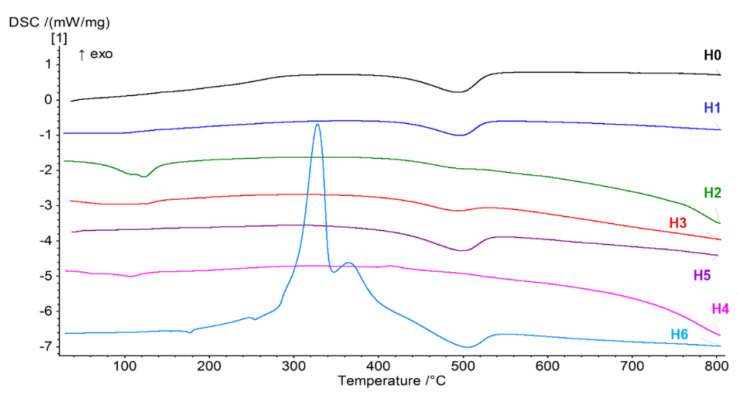
DSC curves of neat halloysite (H0) and halloysite after modifications (H1–H6).

**Figure 7 jfb-14-00167-f007:**
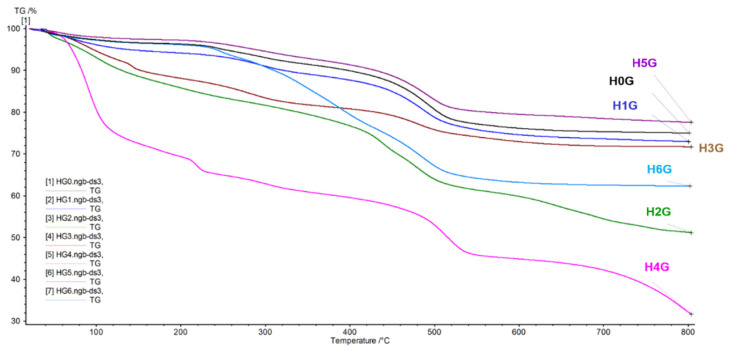
TG curves of halloysite-gentamicin conjugates (H0G–H6G).

**Figure 8 jfb-14-00167-f008:**
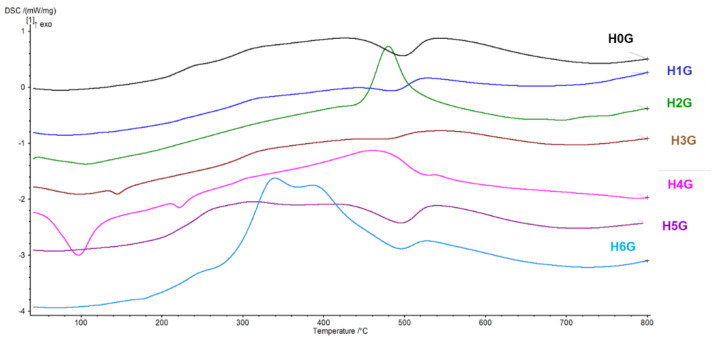
DSC curves of halloysite-gentamicin conjugates (H0G–H6G).

**Figure 9 jfb-14-00167-f009:**
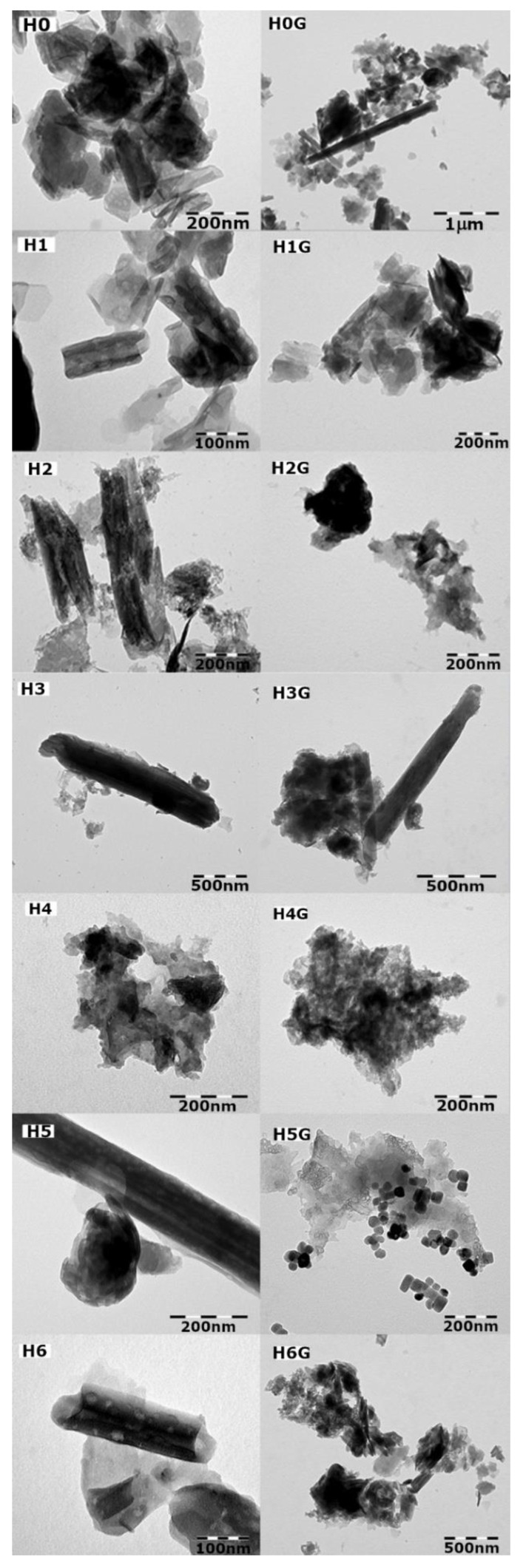
Comparison of TEM microphotographs of neat H0 and modified halloysite H1–H6 with unmodified and modified halloysite after intercalation with the drug H0G–H6G.

**Figure 10 jfb-14-00167-f010:**
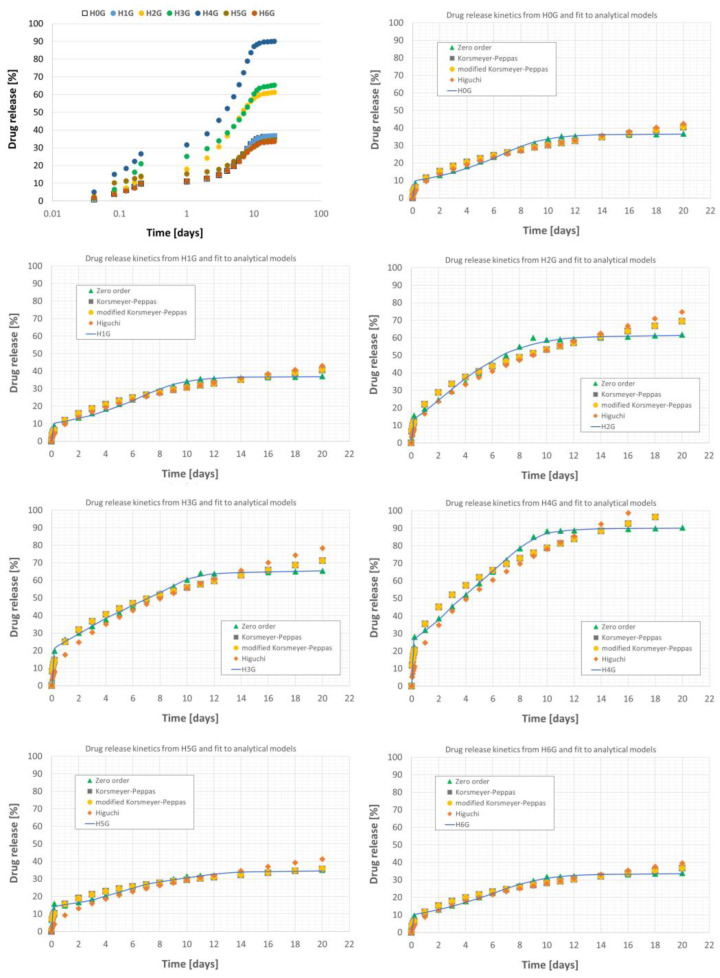
Drug release rates as functions of time for gentamicin intercalated halloysites and their fit to theoretical release models.

**Figure 11 jfb-14-00167-f011:**
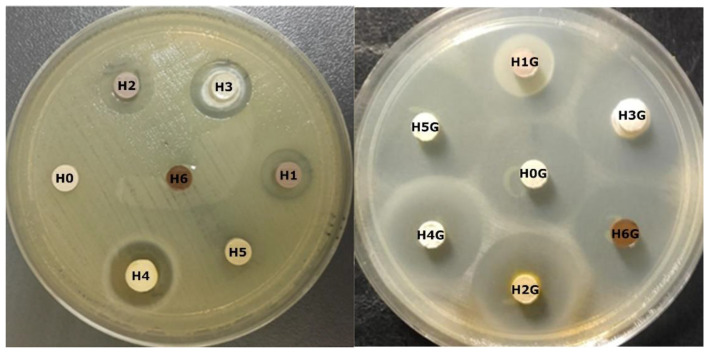
The zones of inhibition of *E. coli* growth for H0–H6 and H0G–H6G samples.

**Table 1 jfb-14-00167-t001:** Tested powder materials before and after modification and intercalation.

Abbr.	Notation	Description
H0	HDU	Dunino halloysite (neat powder)
H1	HDU-H_2_SO_4_	Dunino halloysite modified with sulfuric acid (V)
H2	HDU-eks-H_2_SO_4_	Dunino halloysite (expanded) modified with sulfuric acid (V) in the presence of hydrogen peroxide
H3	HDU-eks-H_3_PO_4_	Dunino halloysite (expanded) modified with phosphoric acid (V) in the presence of hydrogen peroxide
H4	HDU-eks-(NH_4_)_2_S_2_O_8_	Dunino halloysite (expanded) modified with ammonium persulfate in the presence of sulfuric acid (V)
H5	HDU-NaOH	Dunino halloysite modified with sodium hydroxide
H6	HDU-CUR	Dunino halloysite modified with curcumin
H0G	HDU/G	Dunino halloysite + gentamicin (intercalated)
H1G	HDU-H_2_SO_4_/G	Dunino halloysite modified with sulfuric acid (V) + gentamicin (modification + intercalation)
H2G	HDU-eks-H_2_SO_4_/G	Dunino halloysite (expanded) modified with sulfuric acid (V) in the presence of hydrogen peroxide + gentamicin (modification + intercalation)
H3G	HDU-eks-H_3_PO_4_/G	Dunino halloysite (expanded) modified with orthophosphoric acid (V) in the presence of hydrogen peroxide + gentamicin (modification + intercalation)
H4G	HDU-eks-(NH_4_)_2_S_2_O_8_/G	Dunino halloysite (expanded) modified with ammonium persulfate in the presence of sulfuric acid (V) + gentamicin (modification + intercalation)
H5G	HDU-NaOH/G	Dunino halloysite modified with sodium hydroxide + gentamicin (modification + intercalation)
H6G	HDU-CUR/G	Dunino halloysite modified with curcumin + gentamicin (modification + intercalation)
G	G	Gentamicin

**Table 2 jfb-14-00167-t002:** Theoretical and real gentamicin loading efficiency determined for test powders.

Sample	Theoretical Loading Efficiency [%]	Real Loading Efficiency LE [%]
H0G	12.5	5.16
H1G		7.05
H2G		10.91
H3G		10.84
H4G		11.03
H5G		6.75
H6G		5.12

**Table 3 jfb-14-00167-t003:** Parameters of drug release kinetics calculated for all halloysite samples (unmodified and modified) intercalated with gentamicin.

Model	Parameter	Sample						
		H0G	H1G	H2G	H3G	H4G	H5G	H6G
Zero Order (I step)	*f* _0(I)_	0.000	0.000	0.064	0.000	1.134	0.955	0.035
	*K* _0(I)_	0.033	0.035	0.042	0.066	0.090	0.049	0.032
	R^2^	0.991	0.994	0.996	0.992	0.984	0.934	0.998
Zero Order (II step)	*f* _0(II)_	7.793	8.209	14.479	22.580	25.390	12.901	8.343
	*K* _0(II)_	0.002	0.002	0.004	0.003	0.005	0.001	0.002
	R^2^	0.994	0.994	0.991	0.998	0.999	0.996	0.998
Zero Order (III step)	*f* _0(III)_	33.473	33.888	55.855	61.366	86.440	27.786	29.470
	*K* _0(III)_	0.000	0.000	0.000	0.000	0.000	0.000	0.000
	R^2^	0.876	0.876	0.901	0.979	0.864	0.905	0.881
Korsmeyer-Peppas	*K* _k_	0.544	0.618	1.355	1.970	2.865	2.167	0.729
	n	0.419	0.407	0.383	0.349	0.346	0.273	0.382
	R^2^	0.980	0.980	0.986	0.988	0.985	0.979	0.985
Modified Korsmeyer-Peppas	*K* _m_	0.465	0.550	1.356	1.970	2.864	1.786	0.688
	n	0.434	0.418	0.383	0.349	0.346	0.289	0387
	C	0.574	0.385	0.000	0.000	0.000	0.994	0.217
	R^2^	0.978	0.980	0.980	0.988	0.985	0.979	0.985
Higuchi	*K* _h_	0.250	0.253	0.440	0.462	0.650	0.244	0.233
	R^2^	0.979	0.979	0.977	0.978	0.976	0.966	0.981

**Table 4 jfb-14-00167-t004:** Measured zones of *E. coli* growth inhibition around samples not intercalated H0–H6 and intercalated with gentamicin H0G–H6G.

Diameter of *E. coli* Growth Inhibition Zones [mm]			
Not Intercalated		Intercalated with Gentamicin	
H0	7	H0G	24
H1	13	H1G	24
H2	14	H2G	25
H3	15	H3G	27
H4	17	H4G	23
H5	6	H5G	28
H6	6	H6G	22

## Data Availability

The data presented in this study are available on request from the corresponding author.

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
