# Peer review of "Functionalized Halloysite Nanotubes as Potential Drug Carriers"

_jfb, 2023, doi:10.3390/jfb14030167_

Round 1
Reviewer 1 Report
In this research effort a comparative study of halloysite functionalization has been carried out for antimicrobial and sustained release assessment. This work is the extension of previously published robust materials science findings on this functionalized clay type. (https://www.sciencedirect.com/science/article/abs/pii/S0169131717305689; https://www.sciencedirect.com/science/article/abs/pii/S0167577X19302290].
The following changes need to be made to make the current effort more cohesive:
1) The abstract does not have a single quantitative result. Which formulation is superior for release?
2) The introduction needs to be expanded to account for other recent work on hallosyte functionalization.
https://www.hindawi.com/journals/jnm/2022/1068536/
https://www.ncbi.nlm.nih.gov/pmc/articles/PMC6628522/
https://www.ncbi.nlm.nih.gov/pmc/articles/PMC7650711/
3) The material characterization figures, specifically Figure 9 which may be most important link of material science to antimicrobial activity is not clear. Figure 11 the right hand side Figure is not clear. It would be an excellent conclusion to the research effort should the Figure be modified.
4) Quite a few characterization methods have been conducted and illustrated but there is no high level individual summary table (i.e. correlation or lack of crystallinity). Table 4 summarizes results but does not link material properties to crystallinity.
5) The drug release profiles are reported in terms of coefficients. Can the modeled data be superimposed to the actual data (i.e creation of Figure 10b)?
6) Regarding the materials used, in prior art from the group outlined in the first paragraph, it was very clear how and where the materials were procured. The sourcing of the materials needs detail for communication purposes.
Author Response
Manuscript ID: jfb-2284478
Title: Functionalized halloysite nanotubes as potential drug carries
Authors: Ewa Stodolak-Zych *, Alicja Rapacz-Kmita *, Marcin Gajek, Agnieszka Różycka,
Magdalena Dudek, Stanisława Kluska
Journal of Functional Biomaterials
ANSWERS TO REVIEWERS COMMENTS
Thank you very much for your reviews. Below are the responses to the comments of Reviewer 1.
Reviewer #1
In this research effort a comparative study of halloysite functionalization has been carried out for antimicrobial and sustained release assessment. This work is the extension of previously published robust materials science findings on this functionalized clay type. (https://www.sciencedirect.com/science/article/abs/pii/S0169131717305689
https://www.sciencedirect.com/science/article/abs/pii/S0167577X19302290].
The following changes need to be made to make the current effort more cohesive:
1) The abstract does not have a single quantitative result. Which formulation is superior for release?
The abstract has been improved according to the reviewer's suggestions. The changes are highlighted in yellow.
2) The introduction needs to be expanded to account for other recent work on halloysite functionalization.
https://www.hindawi.com/journals/jnm/2022/1068536/
https://www.ncbi.nlm.nih.gov/pmc/articles/PMC6628522/
https://www.ncbi.nlm.nih.gov/pmc/articles/PMC7650711/
Introduction has been changed according to the reviewer's suggestions. The changes are highlighted in yellow.
3) The material characterization figures, specifically Figure 9 which may be most important link of material science to antimicrobial activity is not clear. Figure 11 the right hand side Figure is not clear. It would be an excellent conclusion to the research effort should the Figure be modified.
Figures 9 and 11 have been corrected as suggested by the Reviewer. Revised versions of both figures are included in the manuscript.
4) Quite a few characterization methods have been conducted and illustrated but there is no high level individual summary table (i.e. correlation or lack of crystallinity). Table 4 summarizes results but does not link material properties to crystallinity.
We wish we could use a supplementary method such as the SAED method to retrieve more information on crystallinity, however, at the moment we do not have such research capabilities. We can assess the crystallinity of clay minerals only on the basis of XRD tests and this correlation with the results in Table 4 (antibacterial activity) is difficult to find based on the data we have. SAED imaging would obviously be very helpful, especially since it is currently considered as a novel method for imaging the crystallinity of clay minerals. In the publication, we wanted to perform a series of preliminary tests, primarily aimed at checking the potential of our native halloysite in functionalization processes, as well as in potential applications in drug delivery systems. The crystallinity and its effect on the drug load and release capabilities of halloysite can be, however, investigated in the following research.
5) The drug release profiles are reported in terms of coefficients. Can the modeled data be superimposed to the actual data (i.e creation of Figure 10b)?
Figure 10 has been changed as suggested by the Reviewer. Release kinetics profiles with the fit to all theoretical models for all tested materials were included. Changes have also been included in the description of the figures.
6) Regarding the materials used, in prior art from the group outlined in the first paragraph, it was very clear how and where the materials were procured. The sourcing of the materials needs detail for communication purposes.
The “Materials” section has been supplemented with information on the materials used in the work in accordance with the reviewer's recommendations. The changes are highlighted in yellow.

Reviewer 2 Report
The manuscript is focused on evaluating the influence of halloysite surface modification on its properties and the ability to intercalate with gentamicin. The detailed studies of TEM, XRD, FTIR, and DSC/TG showed the novelty of the surface modification affecting the mineral's stability. After the modification with NaOH and the addition of gentamicin, the hydroxide precipitated, while the expansion with ammonium persulfate destroyed the tubular structures. Modification of the halloysite surface has a significant effect on the amount of gentamicin that can be intercalated and released into the surrounding environment. Antimicrobial studies showed that modifications involving the use of sulfuric acid had a negative effect on the biocidal activity of gentamicin. The largest zones of inhibition of the growth of E. coli bacteria were observed for halloysite modified with NaOH (sample H5G).
The work has been conducted in a good method and detailed in a good method. In addition, some minor corrections can be done to improve the quality of some discussions which are detailed below,
1. Add SAED patterns for each TEM result to clarify the materials' crystallinity and add those sections to the discussion part.
2. Need to clarify the reason for endothermic and exothermic peaks at near 100oC and 480oC for H4G and H2G in figure 8.
Overall, the manuscript is good for the journal after a minor revision.
Author Response
Manuscript ID: jfb-2284478
Title: Functionalized halloysite nanotubes as potential drug carries
Authors: Ewa Stodolak-Zych *, Alicja Rapacz-Kmita *, Marcin Gajek, Agnieszka Różycka,
Magdalena Dudek, Stanisława Kluska
Journal of Functional Biomaterials
ANSWERS TO REVIEWERS COMMENTS
Thank you very much for your reviews. Below are the responses to the comments of Reviewer 2.
The manuscript is focused on evaluating the influence of halloysite surface modification on its properties and the ability to intercalate with gentamicin. The detailed studies of TEM, XRD, FTIR, and DSC/TG showed the novelty of the surface modification affecting the mineral's stability. After the modification with NaOH and the addition of gentamicin, the hydroxide precipitated, while the expansion with ammonium persulfate destroyed the tubular structures. Modification of the halloysite surface has a significant effect on the amount of gentamicin that can be intercalated and released into the surrounding environment. Antimicrobial studies showed that modifications involving the use of sulfuric acid had a negative effect on the biocidal activity of gentamicin. The largest zones of inhibition of the growth of E. coli bacteria were observed for halloysite modified with NaOH (sample H5G).
The work has been conducted in a good method and detailed in a good method. In addition, some minor corrections can be done to improve the quality of some discussions which are detailed below,
- Add SAED patterns for each TEM result to clarify the materials' crystallinity and add those sections to the discussion part.
We are familiar with the SAED method and its possibilities. We wish we could use it, unfortunately, at the moment we do not have such research capabilities. We can assess the crystallinity of clay minerals only on the basis of XRD tests. SAED imaging would obviously be very helpful, especially since it is currently considered a novel method for imaging the crystallinity of clay minerals. In the publication, we wanted to perform a series of preliminary tests, which were primarily aimed at checking the potential of our native halloysite in functionalization processes, as well as in potential applications in drug delivery systems.
- Need to clarify the reason for endothermic and exothermic peaks at near 100oC and 480oC for H4G and H2G in figure 8.
The “Thermal analysis” section has been supplemented with information on DSC analysis for samples intercalated with gentamicin, according to the reviewer's recommendations. The changes are highlighted in yellow.
Overall, the manuscript is good for the journal after a minor revision.

Round 2
Reviewer 1 Report
Thank you for incorporating the suggested changes.
Author Response
Dear Rewiver,
Please find in the attachment the manuscript with revised version. The manuscript has undergone native correction. All changes have been highlighted in yellow.
I hope that the work will now be accepted
Thank you for your work,
Ewa Stodolak-Zych
